# Fine-scale associational effects: Single plant neighbours can alter susceptibility of focal plants to herbivores

Patrick B. Finnerty[1]*, Peter B. Banks[1], Adrian M. Shrader[2], Clare McArthur[1]

1 School of Life & Environmental Sciences, University of Sydney, Sydney, New South Wales, Australia,
2 Department of Zoology and Entomology, University of Pretoria, Pretoria, South Africa

* patrick.finnerty@sydney.edu.au

## Abstract

The neighbourhood of plants in a patch can shape vulnerability of focal plants to herbivores, known as an associational effect. Associational effects of plant neighbourhoods are widely recognised. But whether a single neighbouring plant can exert an associational effect is unknown. Here, we tested if single neighbours indeed do influence the likelihood that a focal plant is visited and eaten by a mammalian herbivore. We then tested whether any refuge effect is strengthened by having more neighbours in direct proximity to a focal plant. We used native plant species and a browser/mixed feeder mammalian herbivore (swamp wallabies (*Wallabia bicolor*)) free-ranging in natural vegetation. We found that a single neighbouring plant did elicit associational effects. Specifically, plant pairs consisting of one high-quality seedling next to a single low-quality plant were visited and browsed by wallabies later and less than pairs of two high-quality seedlings. Having more neighbours did not strengthen these associational effects. Compared with no neighbours, one or five low-quality neighbours had the same effect in delaying time taken for wallabies to first visit a plot and browse on a high-quality focal seedling. While traditionally a 'patch' refers to a broad sphere-of-influence neighbouring plants have on a focal plant, our findings suggest the influence of plant neighbours can range from the nearest individual neighbour to the entire plant neighbourhood. Such fine-scale associational effects are fundamentally important for understanding intricate plant-herbivore interactions, and ecologically important by potentially having knock-on effects on plant survival, in turn influencing plant community structure.

## Introduction

The assemblage of plants in a patch forms a neighbourhood. This plant neighbourhood can influence the susceptibility of an individual plant to being found and eaten by herbivores, known as an associational effect [1–5]. Associational effects are often

**Data availability statement:** The datasets generated during and/or analysed during the current study are available on figshare (https://doi.org/10.6084/m9.figshare.29525723).

**Funding:** Authors CM and PB acknowledge funding from the Australian Research Council (grant number DP190101441).

**Competing interests:** The authors have declared that no competing interests exist.

a consequence of herbivores seeking high-quality 'palatable' patches and avoiding low-quality patches [6–9] to maximise their foraging efficiency. For example, when herbivores select among patches, a focal plant in a high-quality patch is more likely to be visited by a herbivore hence more susceptible to being consumed, known as associational plant susceptibility [10–13]. In contrast, a focal plant in a low-quality patch (e.g., a high density of low nutrient, structurally, and/or chemically defended species) that is avoided by herbivores will be safe from being eaten, known as associational plant refuge [1,2,14,15].

While associational effects of plant patches are widely recognised, the boundaries that define a patch are often nebulous. Generally, 'patch' is used as a short-cut term recognising a sphere-of-influence that neighbouring plants may have on a focal plant [5]. To date, research into the influence of plant patches on focal plant susceptibility to herbivores has largely focused on maximal spheres-of-influence, exploring the effects of the type of plant neighbours and how far they can be from focal plants while still exerting an effect. For example, susceptibility of European beech, *Fagus sylvatica*, to deer browsing decreases with greater distance from, and lower densities of high-quality neighbours (sycamore maple, *Acer pseudoplatanus*) [16]. Similarly, focal heather, *Calluna vulgaris*, are less susceptible to browsing by both red deer, *Cervus elaphus*, and sheep, *Ovis aries*, when further from higher-quality neighbours [12]. However, when considering the influence a patch may have on susceptibility of focal plants to herbivores, the other end of the spectrum has been overlooked: how small can a neighbourhood be, yet still exert an effect?

Selective foraging by herbivores occurs across multiple spatial scales and is hierarchical in nature [17,18]. These spatial scales—from plant communities, to plant neighbourhoods within these communities, to plants within neighbourhoods, and even parts within plants—are effectively nested patches. But the boundaries defining a patch around a focal plant can be ambiguous. A continuum of influence likely exists in shaping focal plant susceptibility to herbivores, from the closest single neighbour to an entire plant neighbourhood. As the number of neighbours in direct proximity to a focal plant increases, associational effects are expected to strengthen, for a range of reasons. More neighbours could change absolute patch quality, convey more information about that quality (e.g., through visual or olfactory cues [19,20]), impose greater costs in terms of search time, and/or provide greater physical or visual obstruction to focal plants from herbivores. Whether single plant neighbours can elicit associational effects, altering the susceptibility of focal plants to herbivores, has been largely explored.

Testing the influence of single plant neighbours on focal plant susceptibility to herbivores has fundamental significance by advancing our understanding of intricate plant-herbivore interactions. Understanding associational effects at these finer spatial scales also has broader ecological significance because, by impacting focal plant survival, plant neighbours may have knock-on effects influencing plant community structure.

Here our **first aim** was to test whether single neighbours can influence if a focal plant is visited and eaten by a mammalian herbivore. In deciding whether to visit or

avoid an area, if herbivores respond to food quality at the scale of a couple of plants, then a single high- or low-quality neighbour should increase or delay time to first visit by a herbivore, influencing a focal plant's vulnerability to being eaten (Fig 1a, 1b, 1c). Our **second aim** was to test whether any refuge effect is strengthened by more, low-quality neighbours in direct proximity to a focal plant (Fig 1a, 1b, 1d). For this aim, we focused on one of our species pairs—the most relevant to associational plant refuge.

In our study, the herbivore we used was a browser/mixed feeder, the swamp wallaby (*Wallabia bicolor*) [21–23], free-ranging in native eucalypt woodland. The swamp wallaby is an Australian macropod (mean body weight female 15 kg, male 19 kg; [24], abundant, and ecologically equivalent to many mammalian herbivores in ecosystems around the world, such as various species of deer in Asia, Europe and America, and antelope and elephant in Africa and Asia. Like these species, swamp wallabies are abundant and a key driver of plant community dynamics [25–27].

To select a low-quality plant species, we conducted haphazard transects in search for a plant that displayed no signs of browsing damage. From these transects we selected *Boronia pinnata*, a small, highly odorous native shrub species. This species has been previously shown to be avoided by swamp wallabies within the study site [28]. At the study site, *B. pinnata* plants generally grow alone, and not in clumps with other *B. pinnata*. We selected two eucalypt species (*Eucalyptus punctata* and *Corymbia gummifera*) as higher-quality plants. These were both a known food source of wallabies at the study site (based on [19] and on unpublished work by [29] and [30]). We confirmed the quality of these plants to wallabies as part of the experiment

## Materials and methods

### Study site

Experiments were run in Ku- ring- gai Chase National Park, New South Wales, Australia (33°41′33″S, 151°08′44″E) from May 2021 to November 2022. All experimental sites were in *Eucalyptus* woodland dominated by scribbly gum (*Eucalyptus haemastoma*), red bloodwood (*Corymbia gummifera*), yellow bloodwood (*Corymbia eximia*), and grey gum (*Eucalyptus punctata*, our focal plant species) with complex shrub and ground vegetation layers. Animal ethics approval was granted

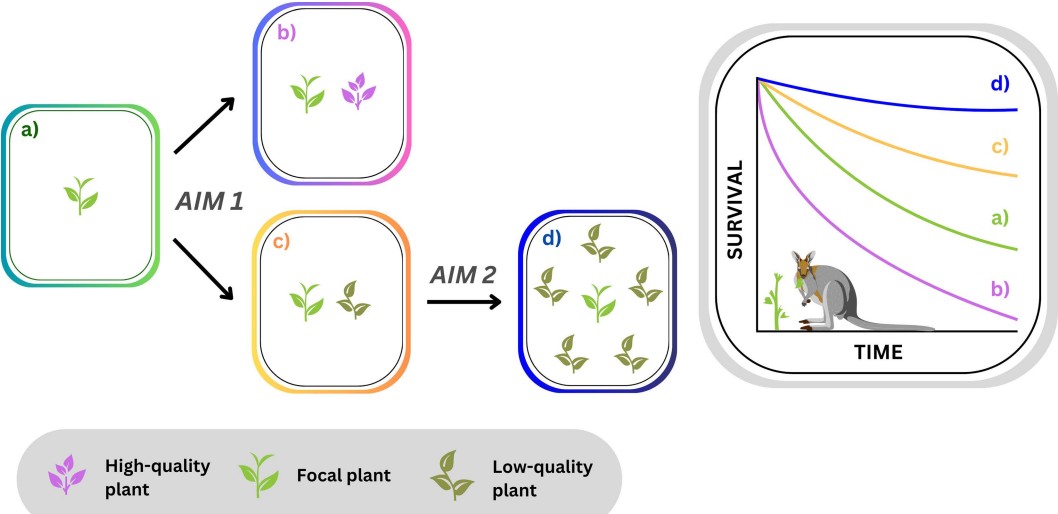

**Fig 1. Predicted associational effects of single or multiple plant neighbours** (a) a single focal plant alone becomes more susceptible to being found and eaten by a herbivore when next to (b) a single high-quality neighbour; less susceptible when next to (c) a single low-quality neighbour; and even less susceptible when (d) surrounded by a neighbourhood of low-quality plants; ultimately affecting survival of the focal plant population over time (RHS).

by the University of Sydney's Animal Ethics Committee (protocol number 2022/2196) and scientific license (SL102186) was obtained from New South Wales Office of Environment and Heritage to undertake the fieldwork.

## Aim 1: Does a single plant neighbour elicit an associational effect?

Using these three species, we tested three paired treatments (Pair, 3 levels) using one plant of each of two species adjacent to one another, and one pair per plot. One treatment paired an *E. punctata* seedling with a *C. gummifera* seedling (n = 17 plots). Another paired an *E. punctata* seedling with a *B. pinnata* plant (n = 23 plots). The third treatment paired a *C. gummifera* seedling with a *B. pinnata* plant (n = 10 plots). The smaller sample size for the *C. gummifera* and *B. pinnata* combination was due to limited availability of *C. gummifera* seedlings. All three pairs were deployed in a completely randomised design at our study site (total n = 50 plots). We placed plots a minimum of 30 m apart, so one plot could not be seen from another. We used naturally occurring *B. pinnata* shrubs growing at the site (375 ± 19 mm). Seedlings of *E. punctata* and *C. gummifera* were of similar height (370 ± 25 mm), planted in soil in tube stock pots (10 cm diam), and supplied by one nursery (Plants Plus Cumberland Forest Nursery, West Pennant Hills, Sydney).

For the treatment pairing *E. punctata* and *C. gummifera*, seedlings were placed 20 cm apart at either end of a clear tub (Fig 2a), allowing easy identification of which species was browsed in video footage For treatments involving *B. pinnata*, the eucalypt seedling was positioned at one end of a clear plastic tub, with the 'free' end of the tub placed next to a *B. pinnata* shrub, again 20 cm apart. A seedling pot with soil was included at the *B. pinnata* end of the tub to account for any effects of open soil (Fig 2b). The side on which each species was presented within each treatment was randomised by coin toss.

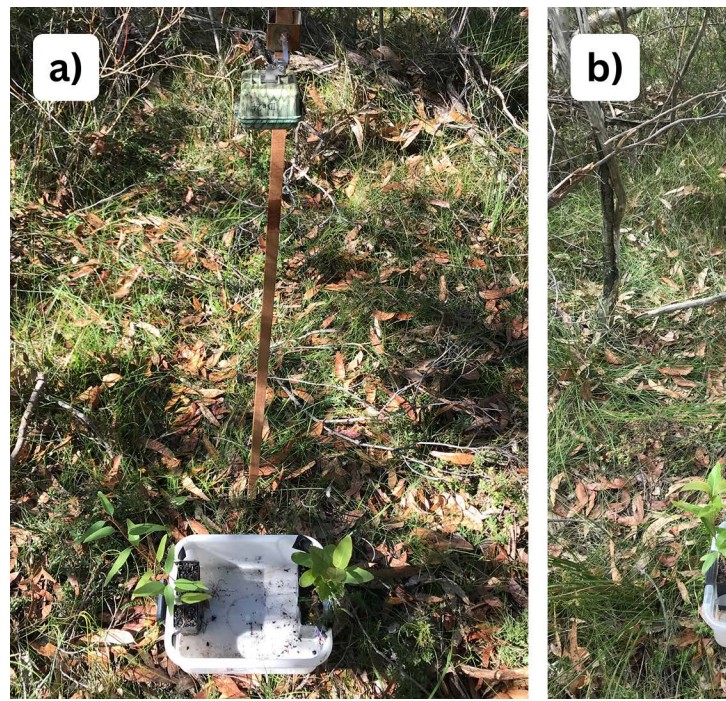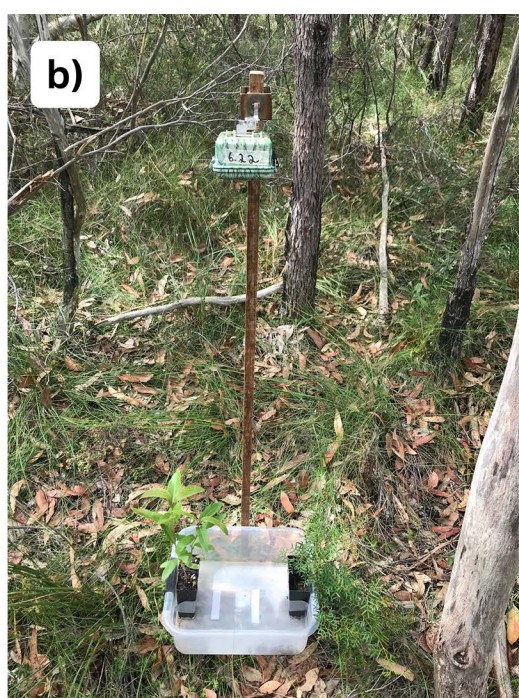

**Fig 2. Aim 1 (single neighbour experiment) images show a motion-triggered camera above a tub holding (a) an *E. punctata* seedling (left) next to one *C. gummifera* seedling (right) and (b) an *E. punctata* seedling (left) next to one naturally occurring *B. pinnata* plus pot with soil (right); the C. gummifera and B. pinnata treatment is not shown.**

At each Pair we set one motion-triggered infra-red trail camera (ScoutGuard SG560K or SG2060-K; Professional Trapping Supplies Pty Ltd, Molendinar, QLD, Australia) fastened to a wooden stake (camera height = 0.7 m) and placed directly over the pair of plants. Cameras were set to record for 1- minute with instant re-trigger. These videos were used to quantify the time to first visit a plot (days) by a wallaby (hereafter *time to 1st visit*), and time to first browse a plant (days) within a plot by a wallaby (hereafter *time to 1st browse*). If browsed, we visually estimated the percentage of foliage consumed (% intervals of 0, 25, 50, 75 and 100%) from each plant as seen on camera.

### Aim 2: Does a neighbourhood of plants elicit stronger associational effects?

Here, we tested whether a refuge effect is strengthened by more low-quality neighbours in direct proximity to a focal plant, using the high-quality *E. punctata* and low-quality *B. pinnata* Pair. All plants, including *B. pinnata*, were in soil in pots (25 cm diam), and obtained from one nursery (Plants Plus Cumberland Forest Nursery, West Pennant Hills, Sydney).

We compared four Treatments (*n* = 16 per Treatment); a single focal *E. punctata* seedling next to a single *B. pinnata* (single neighbour, Fig 3a), a single focal *E. punctata* surrounded by five B. pinnata (neighbourhood, Fig 3b), a single focal *E. punctata* with no neighbours as an untreated control (Fig 3c), and a single focal *E. punctata* surrounded by five pots filled with soil as a procedural control (Fig 3d). All four Treatments were deployed at our study site in plots at least 30 m apart in a completely randomised design (total n = 64 plots). In the single neighbour treatment, *B. pinnata* was placed 50 cm away from the focal *E. punctata*. In the neighbourhood treatment, *B. pinnata* plants, and in the procedural control, pots with soil, were evenly spaced around the focal *E. punctata* at a 50 cm radius. This spacing ensured that *B. pinnata* did not completely visually obscure or physically obstruct wallabies' access to the focal *E. punctata* seedlings.

We calculated a visibility score for the single focal plant per plot, to be used in later analyses as a potential explanation of any observed associational refuge effects. Ordinal scores using visual estimates were used to determine how much of the focal *E. punctata* seedling could be seen from 1 m distance at an average adult swamp wallaby height of 70 cm [31]; 0 = 0%, 1 = 1–25%, 2 = 26–50%, 3 = 51–75%, 4 = 76–100%; where percentages refer to the proportion of the focal *E. punctata* seedling visible. Visibility point scores were taken at 10 evenly spaced locations around each seedling. Point scores were later converted to midpoint values and an average used as a single visibility score.

At each plot we set a motion-triggered infra-red trial camera (ScoutGuard SG560K or SG2060-K; Professional Trapping Supplies Pty Ltd, Molendinar, QLD, Australia). The camera was fastened to a wooden stake (camera height = 0.7 m, distance to seedling = 1.5 m) at an approximate 45° angle towards the plants (Fig 3c and 3d). Cameras were set to record for 1-minute with instant re-trigger. As per the single neighbour trial, these videos were used to quantify the *time to 1st visit* a plot (by a wallaby, in days), and *time to 1st browse* a focal *E. punctata* (by a wallaby, in days) within a plot. If browsed, we estimated the percentage of foliage consumed from the *E. punctata* seedling as for the single neighbour trial. We ran this experiment for 9 weeks, which was the time it took for at least 90% of plots of each Treatment to be visited by wallabies.

### Statistical analysis

**Aim 1: Does a single plant neighbour elicit an associational effect?.** We tested effect of neighbours in three ways: *time to 1st visit* a plot, *time to 1st browse* a *focal plant*, and *% of focal plant browsed*. To test whether there was an effect of Pair on *time to 1st visit* we used a Cox proportional-hazards model ('survival' package [33]) to model 'survival' (where failure is based on *time to 1st visit* a plot) as a function of Pair (3 levels, fixed factor). This model takes into account right-censored data (i.e., if a plot was not visited within the experimental timeframe). Data were censored for any un-visited plots (i.e., plots where no visit occurred during the study). The model was also used to calculate hazard ratios (HR) (exp(coef)) between Pairs for *time to 1st visit* for the three pairwise comparisons.

To test whether there was an effect of neighbour on *time to 1st browse* a focal plant (either *E. punctata* or *C. gummifera*), we used a mixed effects Cox proportional hazard model to model 'survival' (where failure is based on *time to 1st browse*) as a function of neighbour within Pair (fixed factor, 4 levels). There were 4 levels because there were four focal

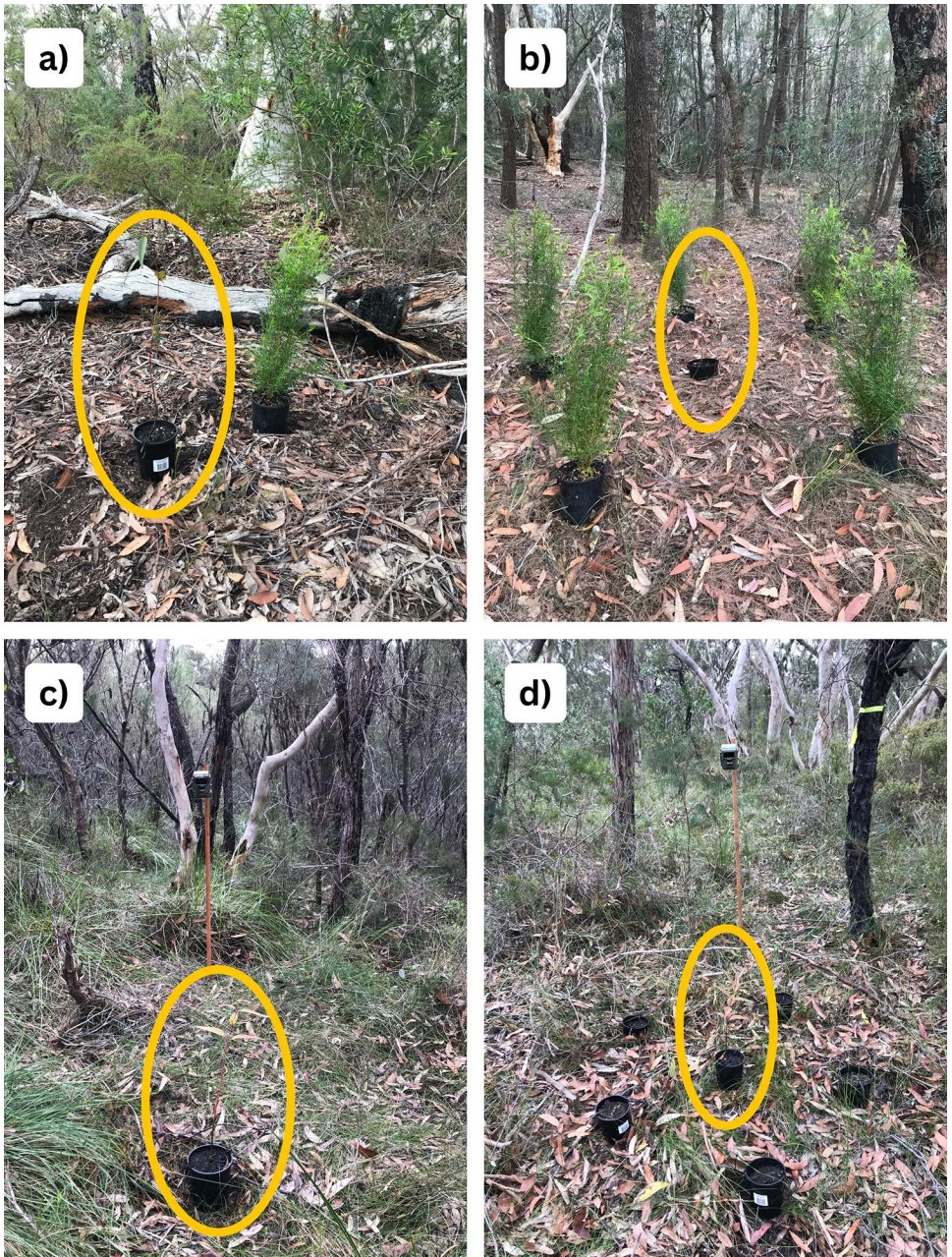

**Fig 3. Aim 2 (neighbourhood experiment) Treatments: (a) a single neighbour of one *B. pinnata* next to one *E. punctata* (left), (b) a neighbour-hood of five *B. pinnata* surrounding one *E. punctata*, (c) an untreated control, one *E. punctata* on its own and (d) a procedural control, five pots with soil surrounding one *E. punctata*.** Image (c) and (d) show experimental set up of motioned-triggered cameras fastened to wooden stakes. Focal plants are circles in yellow.

plant—neighbour combinations: focal *E. punctata* with neighbour *C. gummifera*; focal *C. gummifera* with neighbour *E. punctata*; focal *E. punctata* with neighbour *B. pinnata* (never browsed); and focal *C. gummifera* with neighbour *B. pinnata* (never browsed).We included Plot in the model as a random factor to account for two focal plants (hence possible covariance for browsing) in the *E. punctata*–*C. gummifera* combination

**The model was: Survival (*time to 1st browse focal plant*) ~ neighbour within Pair + (1|Plot).** This model takes into account right-censored data. Data were censored for any plots visited but plants un-browsed. Plots that were not visited were excluded. The model was also used to calculate pairwise hazard ratios between focal species within a Pair.

To compare foliage browsed of focal *E. punctata* between the *E. punctata–C. gummifera* and *E. punctata–B. pinnata* combinations, and on focal *C. gummifera* between the *C. gummifera–E. punctata* and *C. gummifera–B. pinnata* combinations, we ran two beta regressions ('betareg' package [34]) with a logit link function. Percentage browsed data for a given species (*E. punctata* or *C. gummifera*) was included if a plot was visited and at least one plant within the plot was browsed. We excluded data if neither plant was browsed (2 out of 35 valid visits).

To confirm that *B. pinnata* was a low-quality species, and *E. punctata* and *C. gummifera* were higher-quality species, we compared percentage foliage browsed of each species within a Pair at first browsing visit. We ran a paired t-test for each of the three Pairs. Percentage browsed data for both species was included if a plot was visited and at least one plant within the plot was browsed. We excluded data if neither plant was browsed (2 out of 35 valid visits). Data met assumptions for normality and heterogeneity of variance. All statistical analysis was conducted in R (version 4.2.0; R Core Team, 2022)). Data were plotted using 'ggplot2' [32].

**Aim 2: Does a neighbourhood of plants elicit stronger associational effects?.** To test whether there was an effect of Treatment on *time to 1st visit* we used a Cox proportional-hazards model ('survival' package [33]) to model 'survival' (failure as *time to 1st visit* a plot) as a function of Treatment (fixed factor, 4 levels), taking into account right-censored data (i.e., un-visited plots). Pairwise hazard ratios were calculated as for Aim 1.

To test whether there was an effect of Treatment on *time to 1$^{st}$ browse* a focal *E. punctata*, we used a Cox proportional hazard model to model 'survival' (failure as *time to 1$^{st}$ browse*) as a function of Treatment (fixed factor, 4 levels). Plots that were not visited were excluded but data were censored for any plots visited but focal *E. punctata* un-browsed. Pairwise hazard ratios were calculated (as for Aim 1) between Treatments.

To test whether there was a difference in percentage of focal *E.punctata* foliage consumed at first browsing visit between Treatments we used a beta regression as a function of Treatment with a logit link function. We excluded data if the focal plant was visited, but not browsed (3 out of 53 valid visits).

To test for differences in focal *E. punctata* visibility among Treatments we used a one-way ANOVA. Data met assumptions for normality and heterogeneity of variance. If significant, we compared visibility among treatments by performing multiple pairwise comparisons with the Tukey–Kramer adjustment (indicated by alphabetical superscript in figure) ('emmeans' package [35]). All statistical analysis was conducted in R (version 4.2.0; R Core Team, 2022)). Data were plotted using 'ggplot2' [32].

## Results

### Aim 1: Does a single plant neighbour elicit an associational effect?

*Time to 1st visit* differed significantly as a function of Pair (LR $\chi^2_2$= 13.17, *p*=0.001, Fig 4a). Pairwise hazard ratios showed that the combination of *E. punctata* and *C. gummifera* (both high-quality species), was 5.2 times more likely to be visited than the combination of *E. punctata* and *B. pinnata* (z=3.55, *p*<0.001) (high-quality and low-quality species respectively) and 4.2 times more likely to be visited the combination of *C. gummifera* and *B. pinnata* (z=2.46, *p*=0.01) (high-quality and low-quality species respectively) (Table 1). The *E. punctata–B. pinnata* and *C. gummifera–B. pinnata* combinations were equally likely to be visited (z=−0.39, *p*=0.67).

*Time to 1st browse* a focal species differed significantly as a function of neighbour within Pair ($\chi^2_3$= 8.34, *p*=0.04, Fig 4b). Only *E. punctata* and *C. gummifera* (both high-quality species) were considered as focal species because *B. pinnata* (low-quality) was never browsed. Pairwise hazard ratios showed that focal *E. punctata* when next to one *C. gummifera* was 15.9 times more likely to be browsed than if next to one *B. pinnata* (z=2.29, *p*=0.02) (Table 2). Similarly, focal *C.*

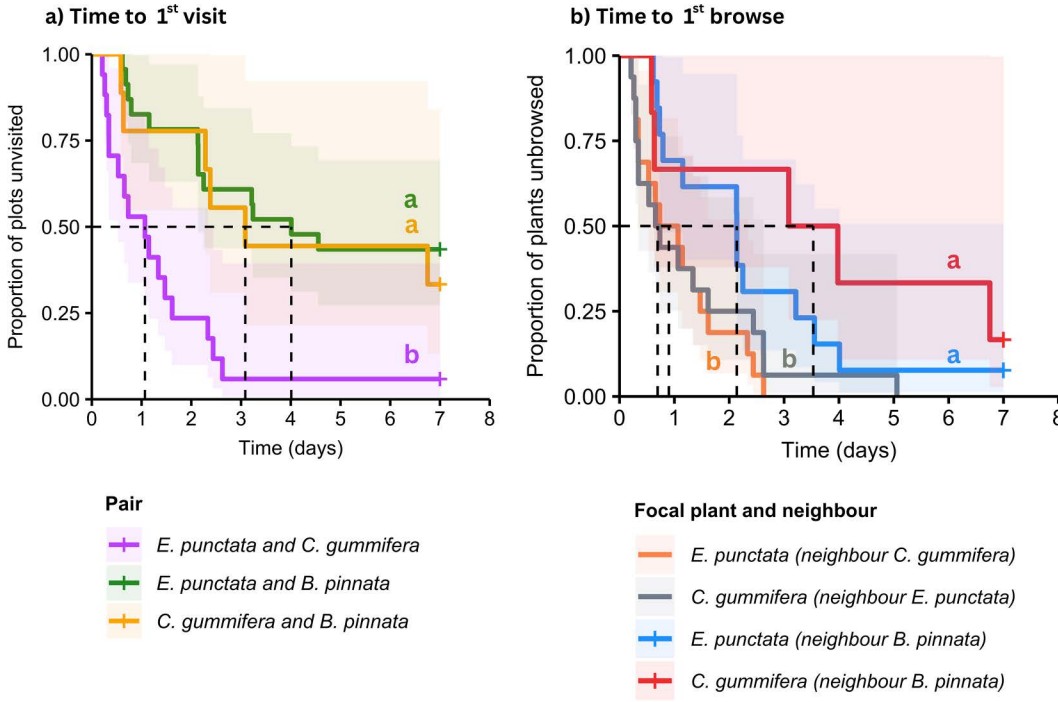

**Fig 4. Survival function (±95% confidence intervals in shaded areas) for (a)** *Time to 1st visit*: survival curves showing the proportion of plots remaining unvisited over 7 days; significant Pair effect (p=0.0003) and (b) *Time to 1st browse*: survival curves showing the proportion of focal plants (noted outside the brackets) remaining unbrowsed over 7 days; significant effect of neighbour within Pair (p<0.001). Dotted lines indicate median survival time for each treatment. Post-hoc pairwise comparisons are shown using alphabetical superscript, where different letters are significantly different.

**Table 1. Pairwise comparisons between** *time to 1st visit* **a Pair. Hazard ratios indicate the likelihood of the first listed Pair being visited compared to the second Pair. If hazard ratios are significantly higher than 1, then the first listed Pair is significantly more likely to be visited than the second.**

| Pairwise comparison | Hazard ratio | z | p-value |
| --- | --- | --- | --- |
| *E. punctata* and *C. gummifera* vs *E. punctata* and *B. pinnata* | **5.17** | **3.55** | **<0.001** |
| *E. punctata* and *C. gummifera* vs *C. gummifera* and *B. pinnata* | **4.15** | **2.46** | **0.01** |
| *E. punctata* and *B. pinnata* vs *C. gummifera* and *B. pinnata* | 0.81 | −0.39 | 0.67 |

**Table 2. Pairwise comparisons between** *time to 1st browse a focal* **plant as a function of neighbour within Pair (noted outside the brackets). Hazard ratios indicate the likelihood that a focal species (in first listed Pair) will be browsed versus another (in second listed Pair). If hazard ratios are significantly higher than 1, than the first listed focal species is significantly more likely to be browsed than the second listed focal species.**

| Pairwise comparison | Hazard ratio | z | p-value |
| --- | --- | --- | --- |
| *E. punctata* (neighbour *C. gummifera*) vs *E. punctata* (neighbour *B. pinnata*) | **15.98** | **2.29** | **0.02** |
| *C. gummifera* (neighbour *E. punctata*) vs *C. gummifera* (neighbour *B. pinnata*) | **35.88** | **2.27** | **0.02** |
| *E. punctata* (neighbour *C. gummifera*) vs *C. gummifera* (neighbour *E. punctata*) | 1.25 | 0.58 | 0.26 |
| *E. punctata* (neighbour *C. gummifera*) vs *C. gummifera* (neighbour *B. pinnata*) | **44.99** | **2.41** | **0.02** |
| *C. gummifera* (neighbour *E. punctata*) vs *E. punctata* (neighbour *B. pinnata*) | **12.74** | **2.11** | **0.03** |
| *E. punctata* (neighbour *B. pinnata*) vs *C. gummifera* (neighbour *B. pinnata*) | 2.81 | 0.65 | 0.51 |

*gummifera* when next to one *E. punctata* was 35.9 times more likely to be browsed than if next to one *B. pinnata* ($z = 2.27$, $p = 0.02$) (Table 2).

When comparing percentage foliage browsed on focal *E. punctata* between Pairs, wallabies consumed significantly more when *E. punctata* was next to *C. gummifera* (another high-quality species) than when it was next to *B. pinnata* (a low-quality species) ($\chi^2_1 = 21.77$, $p < 0.001$, Fig 5). Similarly, when comparing percentage foliage browsed on focal *C. gummifera* between Pairs, wallabies consumed significantly more when *C. gummifera* was next to *E. punctata* (high-quality) than when it was next to *B. pinnata* ( ($\chi^2_1 = 8.52$ $p = 0.004$, Fig 5).

When comparing percentage foliage browsed from both species within a Pair, wallabies browsed significantly more *E. punctata* than *C. gummifera* when these two high-quality species were paired together ($t = -2.64$, $df = 15$, $P = 0.02$, Fig 6a). When *E. punctata* was paired with *B. pinnata* (high vs. low-quality), wallabies browsed significantly more *E. punctata* ($t = -7.37$, $df = 11$, $P < 0.001$, Fig 6b). Similarly, when *C. gummifera* was paired with *B. pinnata* (high vs. low-quality), wallabies browsed significantly more *C. gummifera* ($t = -5.72$, $df = 14$, $P < 0.001$, Fig 6c).

**Aim 2: Does a neighbourhood of plants elicit stronger associational effects?**

*Time to 1st visit* differed significantly as a function of Treatment (LR $\chi^2_3 = 21.65$, $P < 0.001$, Fig 7a). Pairwise hazard ratios show that *E. punctata* (high-quality) with no neighbours (untreated control) were 2.8 times more likely to be visited than *E. punctata* with one *B. pinnata* (low-quality) neighbour ($z = 2.50$, $P = 0.01$), 3.0 times more likely to be visited than *E. punctata* with five *B. pinnata* neighbours ($z = 2.74$, $P = 0.006$). *E. punctata* were equally likely to be visited by wallabies when surrounded by five or one *B. pinnata* neighbour ($z = 0.21$, $P = 0.84$). There was no "pot" effect, as *E. punctata* with no neighbours (untreated control) were equally likely to be visited as *E. punctata* with five pots with soil (procedural control) ($z = 0.22$, $P = 0.59$) (Table 3).

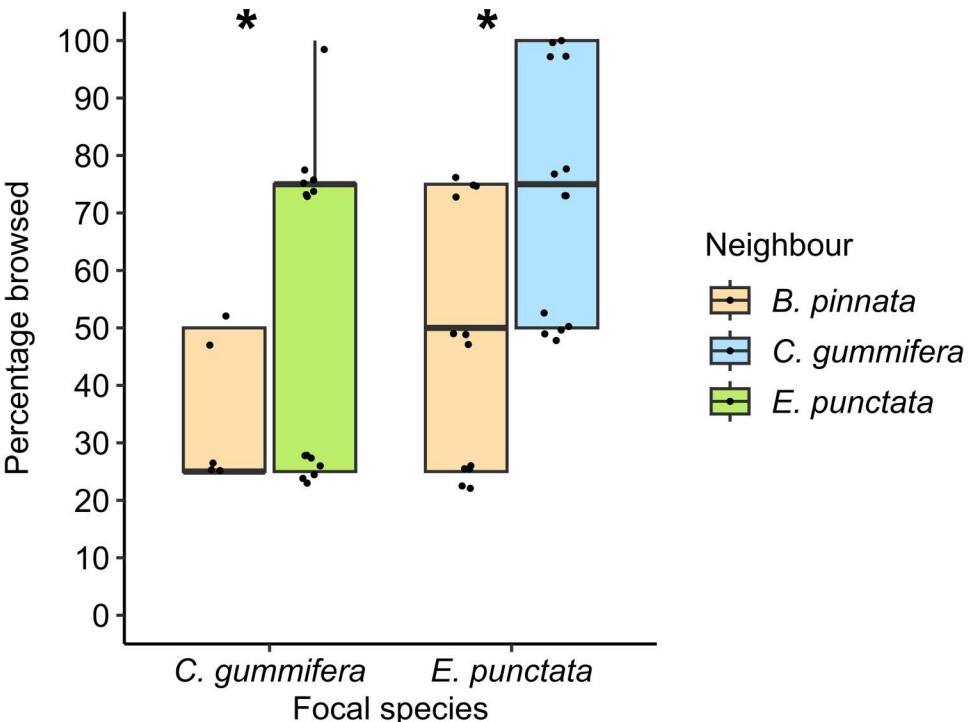

**Fig 5. Comparison of percentage foliage browsed from focal *C. gummifera* and *E. punctata* seedlings (high-quality species) between Pairs at first browsing visit.** Boxplots shows median, 1st and 3rd quartile, and maximum and minimum values (within 1.5×IQR). Raw data points are displayed as jittered full black points. Asterisks indicate significant differences for each focal plant species.

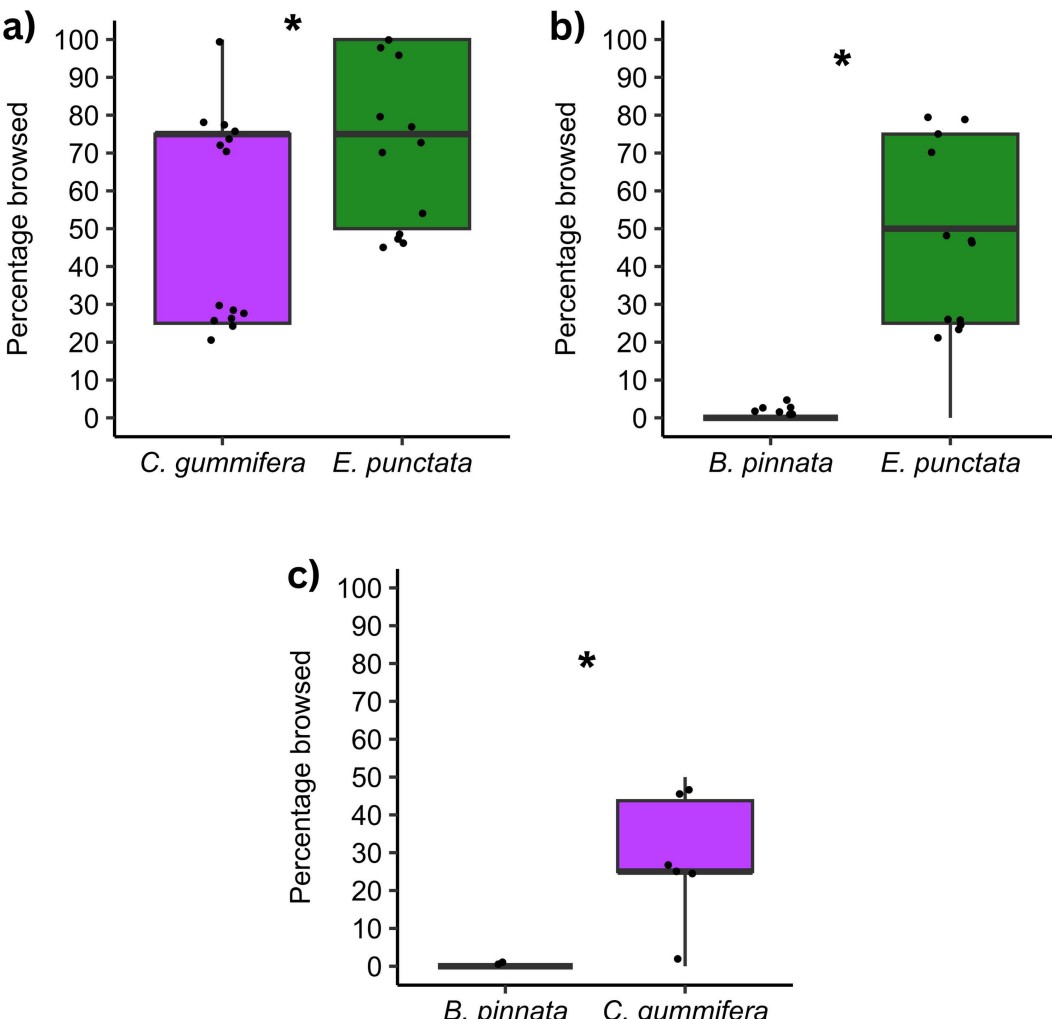

**Fig 6. Percentage foliage browsed from each species within a Pair by wallabies at first browsing visit.** Boxplots shows median, 1st and 3rd quartile, and maximum and minimum values (within 1.5 × IQR). Raw data points are displayed as jittered full black points. Asterisks indicate significant difference. Panel a) shows *E. punctata* and *C. gummifera*, b) shows *E. punctata* and *B. pinnata*, and c) shows *C. gummifera* and *B. pinnata*.

*Time to 1st browse* differed significantly as a function of treatment (LR χ$_3$$^2$ = 21.65, *P* < 0.0001, Fig 7b). Pairwise hazard ratios show that *E. punctata* (high-quality) with no neighbours (untreated control) were 2.5 times more likely to be browsed than *E. punctata* with one *B. pinnata* (low-quality) neighbour (z = 2.16, *P* = 0.03), and 4.5 times more likely to be browsed than *E. punctata* with five *B. pinnata* neighbours (z = 3.40, *P* < 0.001). *E. punctata* were equally likely to be browsed by wallabies when surrounded by one or five *B. pinnata* neighbour (z = 1.34, *P* = 0.18). There was no "pot" effect, as *E. punctata* with no neighbours (untreated control) were equally likely to be browsed as *E. punctata* with one pot with soil (procedural control) (z = 1.22, *P* = 0.39) (Table 4). If browsed, the percentage of *E. punctata* foliage browsed was high (mean 87%) and did not differ between Treatments (LR $\chi_3^2$ = 3.60, *P* = 0.31).

Visibility of focal *E. punctata* differed significantly as a function of Treatment (F$_{3, 57}$ = 29.01, p < 0.0001, Fig 8). Visibility was lowest when surrounded by five *B. pinnata* but the visibility score was still high (mean 83%).

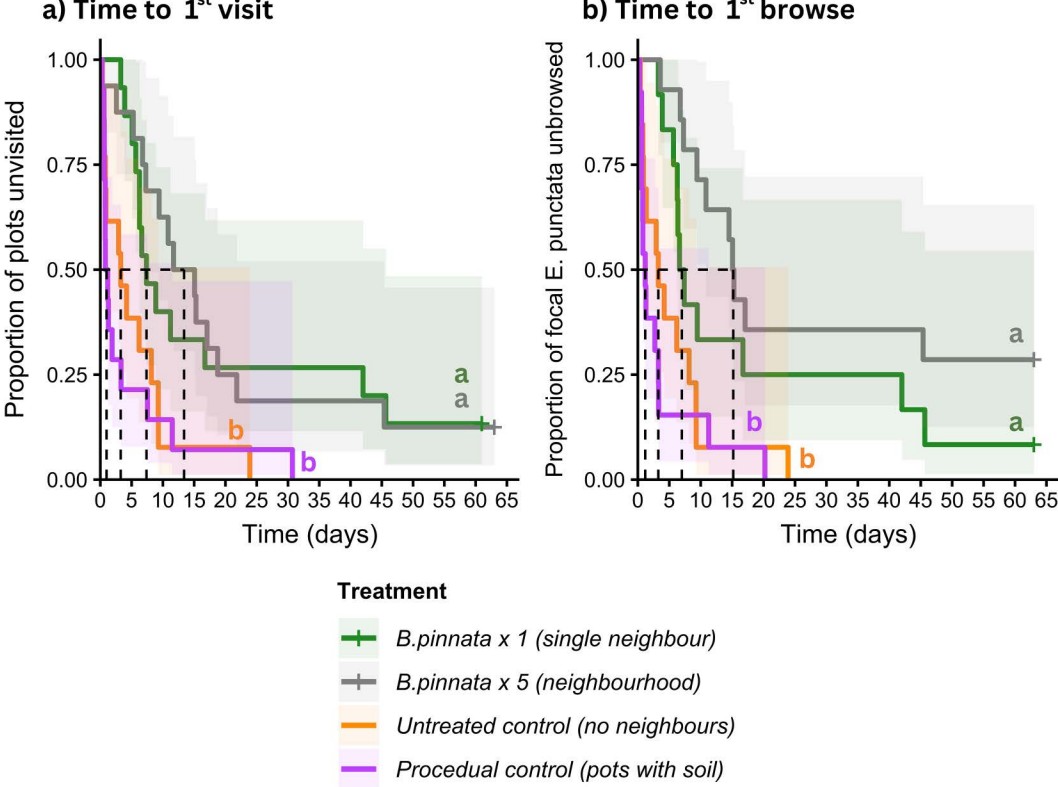

**Fig 7. Survival function (±95% confidence intervals in shaded areas) for (a)** *time to 1st visit*: survival curves showing the proportion of plots remaining unvisited over 9 weeks, significant treatment effect (p < 0.001) and (b) *time to 1st browse*: survival curves showing the proportion of high-quality focal *E. punctata* within plots remaining unbrowsed over 9 weeks, significant treatment effect (p = 0.0001). Dotted lines indicate median survival time for each treatment. Post-hoc pairwise comparisons are shown using alphabetical superscript, where different letters are significantly different.

**Table 3. Pairwise comparisons between** *time to 1*st *visit* a Treatment. Hazard ratios indicate the likelihood of the first listed Treatment being visited compared to the second Treatment. If hazard ratios are significantly higher than 1, than the first listed Treatment is significantly more likely to be visited than the second.

| Pairwise comparison | Hazard ratio | z | p-value |
|---|---|---|---|
| Untreated control (no neighbours) vs *B. pinnata* x 1 (Single neighbour) | **2.75** | **2.50** | **0.01** |
| Untreated control (no neighbours) vs *B. pinnata* × 5 (Neighbourhood) | **2.98** | **2.74** | **0.006** |
| *B. pinnata* x 1 (Single neighbour) vs *B. pinnata* × 5 (Neighbourhood) | 1.08 | 0.21 | 0.84 |
| Untreated control (no neighbours) vs Procedural control (pots with soil) | 1.24 | 0.22 | 0.59 |
| Procedural control (pots with soil) vs *B. pinnata* x 1 (Single neighbour) | **3.40** | **3.06** | **0.002** |
| Procedural control (pots with soil) vs *B. pinnata* × 5 (Neighbourhood) | **3.68** | **3.34** | **< 0.001** |

## Discussion

Our results demonstrate that single neighbours can elicit associational effects. In deciding whether to visit or avoid a pair, wallabies responded to a single plant neighbour, in turn influencing vulnerability of a focal plant to being browsed. Specifically, pairs of one high-quality *E. punctata* or *C. gummifera* seedling next to a single low-quality *B. pinnata* plant were visited and subsequently browsed later by wallabies than a pair of *E. punctata* and *C. g ummifera* seedlings. In delaying time

**Table 4. Pairwise comparisons between *time to 1*st *browse* a focal *E. punctata* as a function of Treatment. Hazard ratios indicate the likelihood of focal *E. punctata* being browsed within the first listed Treatment compared to *E. punctata* being browsed within the second listed Treatment. If hazard ratios are significantly higher than 1, than a *E. punctata* within the first listed Treatment is significantly more likely to be browsed than a *E. punctata* within the second listed Treatment.**

| Pairwise comparison | Hazard ratio | z | p-value |
|---|---|---|---|
| Untreated control (no neighbours) vs *B. pinnata* x 1 (Single neighbour) | **2.49** | **2.16** | **0.03** |
| Untreated control (no neighbours) vs *B. pinnata* × 5 (Neighbourhood) | **4.49** | **3.40** | **< 0.001** |
| *B. pinnata* x 1 (Single neighbour) vs *B. pinnata* × 5 (Neighbourhood) | 1.80 | 1.34 | 0.18 |
| Untreated control (no neighbours) vs Procedural control (pots with soil) | 1.40 | 1.22 | 0.39 |
| Procedural control (pots with soil) vs *B. pinnata* x 1 (Single neighbour) | **3.61** | **2.95** | **0.003** |
| Procedural control (pots with soil) vs *B. pinnata* × 5 (Neighbourhood) | **6.31** | **4.16** | **< 0.001** |

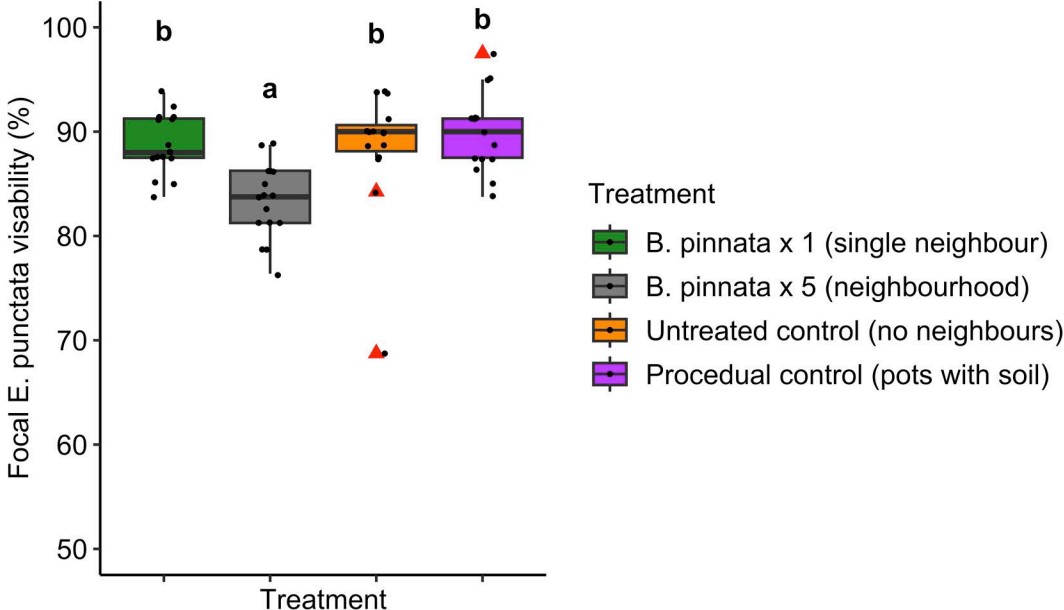

**Fig 8. Boxplots of average percentage visibility of focal *E. punctata* within each Treatment.** Each boxplot shows median, 1st and 3rd quartile, and maximum and minimum values (within 1.5 × IQR). Post-hoc pairwise comparisons are shown using alphabetical superscript, where different letters are significantly different. Red triangles are outliers. Raw data points are displayed as jittered full black points. Y axis begins at 50% focal seedling visibility.

to first visit a pair, a single low-quality neighbour influences herbivore behaviour at a crucial, early stage in the foraging process, when investment is low and hence easier to alter [36,37]. Our results also show that later stages of the foraging process were shaped by single low-quality neighbours, with time to first browse delayed and amount eaten lower for both *E. punctata* and *C. gummifera* seedlings when next to *B. pinnata* then when next to each other. These results are the first to demonstrate associational refuge effects of neighbouring plants occurring at such fine—single neighbour—spatial scales.

Contrary to our predictions (from Fig 1), associational refuge effects were not strengthened by having more low-quality neighbours. Specifically, the delay in wallabies visiting a plot and subsequently browsing a focal *E. punctata* seedling was equivalent whether it was next to a single *B. pinnata* plant or surrounded by a neighbourhood of five plants. Given

wallabies can use odour cues from afar to judge plant quality [28,38–41] and quantity [19], it is perhaps not surprising that the slight reduction in visibility of *E. punctata* caused by five rather than one *B. pinnata* had no impact on foraging. Nevertheless, although a single *B. pinnata* plant may have the same effect on wallabies' ultimate foraging choice as five plants, having five *B. pinnata* plants would likely emit a stronger quantitative odour cue that could be detected from greater distances, potentially influencing how far away wallabies make their foraging decisions, rather than changing their choice itself, or possibly masking the odour of focal plants altogether. Regardless, here, wallabies seem to have avoided *B. pinnata* (hence adjacent *E. punctata*) irrespective of quantity. This implies that at finer spatial scales, the quantity of low-quality neighbours may not be as important in shaping wallaby, and possibly other mammalian herbivore foraging decisions than at larger spatial scales, where the quantity and proximity of low-quality neighbours does matter in influencing associational effect strength [16].

For the single neighbour experiment (Aim 1, which tested whether a single plant neighbour elicits an associational effect), high-quality focal plants were browsed less by wallabies when next to a single low-quality neighbour. In contrast, in the neighbourhood experiment (Aim 2, which tested whether multiple neighbours elicit stronger associational effects), if browsed, high-quality focal *E. punctata* were almost always completely consumed regardless of Treatment. This difference in browsing severity may reflect a difference in focal *E. punctata* quality [8,42] and/or a difference in *B.pinnata* quality between the two experiemnts. Although from the same nursery, *E. punctata* seedlings were purchased at different times within a year and possibly grown under different conditions. Similarly, *B. pinnata* was naturally occurring in Aim 1 but nursery-grown in Aim 2. Different abiotic growing conditions could affect nutrient and plant secondary metabolite concentrations (e.g. [43]).

Our study highlights that individual plant neighbours can indeed exert significant associational effects, comparable in strength to those observed from a neighbourhood of plants. In demonstrating the influence of single plant neighbours on herbivore foraging decisions, our study reveals a nuanced plant-herbivore interaction of fundamental importance. While traditionally a 'patch' refers to a broad sphere-of-influence neighbouring plants have on a focal plant, our findings suggest the influence of plant neighbours ranges from the nearest individual neighbour to the entire plant neighbourhood.

These subtle and more localised associational effects could have important ecological implications. Selective browsing by herbivores can play a pivotal role in shaping ecosystems [44,45]. Thus, by impacting focal plant susceptibility to herbivores, even single plant neighbours may have knock-on effects influencing plant community structure, and indirectly impact other organisms in these communities via a trophic cascade [46–48].

## Supporting information

**S1 File. Supplementary data and code files.**
(ZIP)

## Author contributions

**Conceptualization:** Patrick B. Finnerty, Peter B Banks, Adrian M Shrader, Clare McArthur.

**Data curation:** Patrick B. Finnerty, Clare McArthur.

**Formal analysis:** Patrick B. Finnerty, Clare McArthur.

**Funding acquisition:** Peter B Banks, Clare McArthur.

**Investigation:** Patrick B. Finnerty.

**Methodology:** Patrick B. Finnerty, Clare McArthur.

**Supervision:** Peter B Banks, Adrian M Shrader, Clare McArthur.

**Visualization:** Patrick B. Finnerty.

**Writing – original draft:** Patrick B. Finnerty.

**Writing – review & editing:** Patrick B. Finnerty, Peter B Banks, Adrian M Shrader, Clare McArthur.

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
