## [Decision Letter · Decision Letter 0]

26 May 2025

PONE-D-25-16194Fine-scale associational effects: single plant neighbours can alter susceptibility of focal plants to herbivoresPLOS ONE

Dear Dr. Finnerty,

Thank you for submitting your manuscript to PLOS ONE. After careful consideration, we feel that it has merit but does not fully meet PLOS ONE’s publication criteria as it currently stands. Therefore, we invite you to submit a revised version of the manuscript that addresses the points raised during the review process.

Both reviewers offer minor suggestions to improve clarity and coherence, such as refining the introduction by including plant descriptions earlier, adjusting figure legends for clarity, and considering alternative explanations for results based on environmental factors. Additionally, one reviewer recommends replacing numerical notation (e.g., (1), (2)) with clearer wording, while the other suggests defining key terms like "neighborhood" earlier and ensuring Aims are referred to by their specific content rather than labels. Overall, the feedback is highly positive, with both reviewers commending the novelty of the study and its contribution to the field.

Please submit your revised manuscript by Finnerty. If you will need more time than this to complete your revisions, please reply to this message or contact the journal office at plosone@plos.org. Please include the following items when submitting your revised manuscript:

We look forward to receiving your revised manuscript.

Kind regards,

Showkat Ahmad Ganie, Ph.D.

Academic Editor

PLOS ONE

Journal Requirements:

3. Thank you for stating the following financial disclosure: [Authors CM and PB acknowledge funding from the Australian Research Council (grant number DP190101441).]. 

4. Thank you for stating the following in the Acknowledgments Section of your manuscript: [C.M. and P.B.B. acknowledge funding from the Australian Research Council (grant number DP190101441).]

Please remove any funding-related text from the manuscript and let us know how you would like to update your Funding Statement. Currently, your Funding Statement reads as follows: [Authors CM and PB acknowledge funding from the Australian Research Council (grant number DP190101441).]. 

Additional Editor Comments:

Reviewers offer minor suggestions to improve clarity and coherence, such as refining the introduction by including plant descriptions earlier, adjusting figure legends for clarity, and considering alternative explanations for results based on environmental factors. Additionally, one reviewer recommends replacing numerical notation (e.g., (1), (2)) with clearer wording, while the other suggests defining key terms like "neighborhood" earlier and ensuring Aims are referred to by their specific content rather than labels. Overall, the feedback is highly positive, with both reviewers commending the novelty of the study and its contribution to the field.

Reviewers' comments:

Reviewer's Responses to Questions

**Comments to the Author**

1. Is the manuscript technically sound, and do the data support the conclusions?

Reviewer #1: Yes

Reviewer #2: Yes

2. Has the statistical analysis been performed appropriately and rigorously? 

Reviewer #1: Yes

Reviewer #2: Yes

3. Have the authors made all data underlying the findings in their manuscript fully available?

Reviewer #1: Yes

Reviewer #2: Yes

4. Is the manuscript presented in an intelligible fashion and written in standard English?

Reviewer #1: Yes

Reviewer #2: Yes

5. Review Comments to the Author

Reviewer #1: Review of PONE-D-25-16194 “Fine-scale associational effects: single plant neighbours can alter susceptibility of focal plants to herbivores”

The authors describe a well-designed study that determined that a single unpalatable plant can reduce visitation and browsing of a palatable plant. The methodology and analysis of the experiment are sound and the results are both important and interesting. I just have a few minor comments that I hope will help with the clarity of the manuscript.

Introduction: Lines 95-100 The herbivore used for the study, the swamp wallaby, is described but not the plants used. A sentence or two about your three plants selected for the study would be helpful here. That information is provided in material and methods so could just be moved into the introduction.

Materials and Methods: Lines 124-127: Naturally occurring B. pinnata shrubs were used, while nursery grown E. punctata and C. gummifera were used. You state the size of the nursery grown plants were of similar size but did not give any description of the size of the B. pinnata plants chosen. Did you choose B. pinnata plants of similar size? If you did please include that information in methods. If similarly sized B. pinnata were not chosen that needs to be addressed as a possible limitation to the study.

Material and methods, results, and figures: For Aim 1 and 2, you use the term Pair or Treatment and then (1),(2) etc to denote your different Pairs or Treatments. The (#) is often distracting and makes the number seem important rather than just denoting the different pairs or treatments. I suggest removing the () around your numbers. It is especially distracting in the figure legends since there are (a), (b) etc to denote which panel is being described.

Figures: Figure 6 legend: lines 294-297 need to describe what panels (a), (b), and (c) denote.

Discussion: Lines 375-387: Reasons for why more of the focal plant E. punctata was foraged in Aim 2 versus Aim 1 are discussed here. Is it also possible that the reason more E. punctate was foraged on in Aim 2 is that the plots seem to have less undergrowth/browse nearby than the plots in Aim 1. I am basing this off the pictures of the experimental setups in Figures 2 and 3.

Reviewer #2: This is an excellent manuscript that sheds new light on the understudied question of how small a plant’s neighborhood can be while still influencing its susceptibility to herbivores. By shifting focus to the smallest spatial scale at which neighborhood effects operate, the authors provide a novel perspective on plant–herbivore interactions. This manuscript was well-written, the experimental design and statistical approaches were robust, and the interpretations of the data appropriate. This manuscript was a delight to review. I have a few general, but minor comments that I think would strengthen the manuscript.

A general suggestion: This manuscript has a quite a few moving parts with multiple plant species and combinations and numbers of said species. Given this, it can be hard to keep track of what “Aim 1” vs. “Aim 2” when they are used in a sentence out of context (e.g. L376). While I think it is fine to name them Aim 1 and Aim 2, I suggest referring to them based on their content (i.e., what they are actually testing).

The opening paragraph is well-written, however, I think a few tweaks to it might strengthen the foundational ideas for the rest of the manuscript. For example:

L43: I think it might be helpful to define a neighborhood (in your context) here, given that it is a term heavily used through the manuscript. For example, it might be useful to say something along the lines of “The assemblage of plants in a patch forms a neighborhood, which can…”

L46: I suggest adding an additional sentence between the ones that start and end on L46 to set up the idea that the neighborhood can either have a positive or negative effect on a given plant. The sentence beginning with 'When herbivores…' could be revised to start with 'For example, when herbivores…,' which would create a smoother transition and more clearly set up the contrast with the following sentence that begins 'In contrast…,' highlighting the opposing effects of a neighborhood.

In the Statistical Analysis section:

L185: Is it possible to include this information elsewhere to omit an orphan paragraph?

In the Discussion:

L368-374: Might it be possible that more plants could emit a strong smell that might be detectable from larger distances? Ultimately, this could influence the distance at which decisions could be made rather than the choice itself.

6. PLOS authors have the option to publish the peer review history of their article (what does this mean?). If published, this will include your full peer review and any attached files.

Reviewer #1: No

Reviewer #2: No

---

## [Author Response · Author response to Decision Letter 1]

9 Jul 2025

Response to reviewers – PLOS One PONE-D-25-16194

“Fine-scale associational effects: single plant neighbours can alter susceptibility of focal plants to herbivores”

We thank the two reviewers for providing comment on our manuscript and suggestions for improvement. We have revised the manuscript in response to the reviewers’ queries and suggestions. We have listed every comment raised by each reviewer below, with a statement of how and where the manuscript has been amended in response to these comments, or our justification for making no change. All line numbers referenced below refer to the clean version of the manuscript uploaded.

Journal Requirements:

RESPONSE: The manuscript meets PLOS ONE’s style requirements.

2. In your Methods section, please provide additional information regarding the permits you obtained for the work. Please ensure you have included the full name of the authority that approved the field site access.

RESPONSE: Additional information regarding the permits you obtained for the work have been provided.

RESPONSE: L116 – 119: Animal ethics approval was granted by the University of Sydney’s Animal Ethics Committee (protocol number 2022/2196) and scientific license (SL 102186) were obtained from New South Wales Office of Environment and Heritage to undertake the fieldwork.

3. Thank you for stating the following financial disclosure: [Authors CM and PB acknowledge funding from the Australian Research Council (grant number DP190101441).].

RESPONSE: The financial disclosure statement has been removed from the manuscript as per comment below. However in the cover letter and the online Funding Statement section we have amended our wording as such:

C.M. and P.B.B. acknowledge funding from the Australian Research Council (grant number DP190101441). The funders had no role in study design, data collection and analysis, decision to publish, or preparation of the manuscript.

RESPONSE: As above.

4. Thank you for stating the following in the Acknowledgments Section of your manuscript: [C.M. and P.B.B. acknowledge funding from the Australian Research Council (grant number DP190101441).]

RESPONSE: As above.

Please remove any funding-related text from the manuscript and let us know how you would like to update your Funding Statement. Currently, your Funding Statement reads as follows: [Authors CM and PB acknowledge funding from the Australian Research Council (grant number DP190101441).].

RESPONSE: As above.

RESPONSE: The datasets generated during and/or analysed during the current study are currently available in the Sydney eScholarship Repository - https://hdl.handle.net/2123/33752.

RESPONSE: Additional information regarding the permits you obtained for the work have been provided.

L116 – 119: Animal ethics approval was granted by the University of Sydney’s Animal Ethics Committee (protocol number 2022/2196) and scientific license (SL 102186) were obtained from New South Wales Office of Environment and Heritage to undertake the fieldwork.

RESPONSE: Supporting information has been removed as was R code and data in the oringally submitted manuscript. The datasets generated during and/or analysed during the current study are now currently available in the Sydney eScholarship Repository and hence no longer required as supplementary material - https://hdl.handle.net/2123/33752.

RESPONSE: Reference list is complete and correct.

Editor’s comments

Reviewers offer minor suggestions to improve clarity and coherence, such as refining the introduction by including plant descriptions earlier, adjusting figure legends for clarity, and considering alternative explanations for results based on environmental factors. Additionally, one reviewer recommends replacing numerical notation (e.g., (1), (2)) with clearer wording, while the other suggests defining key terms like "neighborhood" earlier and ensuring Aims are referred to by their specific content rather than labels. Overall, the feedback is highly positive, with both reviewers commending the novelty of the study and its contribution to the field.

RESPONSE: We agree with the reviewers’ comments and have gone through and ensured all minor suggested changes have been addressed in the revised manuscript.

Reviewer #1

The authors describe a well-designed study that determined that a single unpalatable plant can reduce visitation and browsing of a palatable plant. The methodology and analysis of the experiment are sound and the results are both important and interesting. I just have a few minor comments that I hope will help with the clarity of the manuscript.

RESPONSE: We appreciate the reviewer’s comments and suggested corrections. All suggested edits have been included in the revised manuscript.

Introduction: Lines 95-100 The herbivore used for the study, the swamp wallaby, is described but not the plants used. A sentence or two about your three plants selected for the study would be helpful here. That information is provided in material and methods so could just be moved into the introduction.

RESPONSE: As suggested, we have now moved the information provided on the three plant species from the materials and methods section into the introduction.

L101-108: To select a low-quality plant species, we conducted haphazard transects in search for a plant that displayed no signs of browsing damage. From these transects we selected Boronia pinnata, a small, highly odorous native shrub species. This species has been previously shown to be avoided by swamp wallabies within the study site [28]. At the study site, B. pinnata plants generally grow alone, and not in clumps with other B. pinnata. We selected two eucalypt species (Eucalyptus punctata and Corymbia gummifera) as higher-quality plants. These were both a known food source of wallabies at the study site (based on [19] and on unpublished work by [29] and [30]). We confirmed the quality of these plants to wallabies as part of the experiment

Materials and Methods: Lines 124-127: Naturally occurring B. pinnata shrubs were used, while nursery grown E. punctata and C. gummifera were used. You state the size of the nursery grown plants were of similar size but did not give any description of the size of the B. pinnata plants chosen. Did you choose B. pinnata plants of similar size? If you did please include that information in methods. If similarly sized B. pinnata were not chosen that needs to be addressed as a possible limitation to the study.

RESPONSE: Thank you for pointing this out. We did record naturally occurring B. pinnata size, and yes, they were of similar size to the nursery grown E. punctata and C. gummifera we used. We have now included naturally occurring B. pinnata size in the revised manuscript (L129).

Material and methods, results, and figures: For Aim 1 and 2, you use the term Pair or Treatment and then (1),(2) etc to denote your different Pairs or Treatments. The (#) is often distracting and makes the number seem important rather than just denoting the different pairs or treatments. I suggest removing the () around your numbers. It is especially distracting in the figure legends since there are (a), (b) etc to denote which panel is being described.

RESPONSE: Thank you for this suggestion. We agree that the notation was distracting and have now removed all “Pair (1) etc.” and “Treatment (1) etc.” notation throughout the manuscript, including in the Materials and Methods, Results, and figure legends, to aid simplicity and readability.

Figures: Figure 6 legend: lines 294-297 need to describe what panels (a), (b), and (c) denote.

RESPONSE: Included

L306-307: Panel a) shows E. punctata and C. gummifera, b) shows E. punctata and B. pinnata, and c) shows C. gummifera and B. pinnata.

Discussion: Lines 375-387: Reasons for why more of the focal plant E. punctata was foraged in Aim 2 versus Aim 1 are discussed here. Is it also possible that the reason more E. punctate was foraged on in Aim 2 is that the plots seem to have less undergrowth/browse nearby than the plots in Aim 1. I am basing this off the pictures of the experimental setups in Figures 2 and 3.

RESPONSE: Thanks for pointing this out. Aims 1 and 2 were conducted within the same study site, where undergrowth was relatively consistent across all plots. While the photographs in Figures 2 and 3 suggest that there may have been less undergrowth in the plots shown for Aim 2, this is not representative of broader site conditions. We agree this is a valid potential explanation; however, given that undergrowth was similar across the entire site, we do not expect this to have influenced the amount of E. punctata browsing observed between Aims 1 and 2. Unfortunately, we did not take additional photos that better capture the undergrowth in the Aim 2 plots to clarify this point.

Reviewer #2

This is an excellent manuscript that sheds new light on the understudied question of how small a plant’s neighborhood can be while still influencing its susceptibility to herbivores. By shifting focus to the smallest spatial scale at which neighborhood effects operate, the authors provide a novel perspective on plant–herbivore interactions. This manuscript was well-written, the experimental design and statistical approaches were robust, and the interpretations of the data appropriate. This manuscript was a delight to review. I have a few general, but minor comments that I think would strengthen the manuscript.

RESPONSE: We appreciate the reviewer’s comments and suggested corrections. All suggested edits have been included in the revised manuscript.

A general suggestion: This manuscript has a quite a few moving parts with multiple plant species and combinations and numbers of said species. Given this, it can be hard to keep track of what “Aim 1” vs. “Aim 2” when they are used in a sentence out of context (e.g. L376). While I think it is fine to name them Aim 1 and Aim 2, I suggest referring to them based on their content (i.e., what they are actually testing).

RESPONSE: Thank you for this helpful suggestion. We have revised the manuscript to refer to each aim by its content (i.e., the specific experiment being tested) rather than solely as “Aim 1” or “Aim 2” throughout, to improve clarity for readers.

L140-143: Aim 1 (single neighbour experiment) images show a motion-triggered camera above a tub holding (a) an E. punctata seedling (left) next to one C. gummifera seedling (right) and (b) an E. punctata seedling (left) next to one naturally occurring B. pinnata plus pot with soil (right); the C. gummifera and B. pinnata treatment is not shown.

L168 – 172: Aim 2 (neighbourhood experiment) Treatments: (a) a single neighbour of one B. pinnata next to one E. punctata (left), (b) a neighbourhood of five B. pinnata surrounding one E. punctata, (c) an untreated control, one E. punctata on its own and (d) a procedural control, five pots with soil surrounding one E. punctata. Image (c) and (d) show experimental set up of motioned-triggered cameras fastened to wooden stakes. Focal plants are circles in yellow.

L388-397: For the single neighbour experiment (Aim 1, which tested whether a single plant neighbour elicits an associational effect), high-quality focal plants were browsed less by wallabies when next to a single low-quality neighbour. In contrast, in the neighbourhood experiment (Aim 2, which tested whether multiple neighbours elicit stronger associational effects), if browsed, high-quality focal E. punctata were almost always completely consumed regardless of Treatment. This difference in browsing severity may reflect a difference in focal E. punctata quality [8, 40] and/or a difference in B.pinnata quality between the two experiemnts. Although from the same nursery, E. punctata seedlings were purchased at different times within a year and possibly grown under different conditions. Similarly, B. pinnata was naturally occurring in Aim 1 but nursery-grown in Aim 2. Different abiotic growing conditions could affect nutrient and plant secondary metabolite concentrations (e.g. [43]).

The opening paragraph is well-written, however, I think a few tweaks to it might strengthen the foundational ideas for the rest of the manuscript. For example:

L43: I think it might be helpful to define a neighborhood (in your context) here, given that it is a term heavily used through the manuscript. For example, it might be useful to say something along the lines of “The assemblage of plants in a patch forms a neighborhood, which can…”

RESPONSE: Noted and updated.

L43-45: The assemblage of plants in a patch forms a neighbourhood. This plant neighbourhood can influence the susceptibility of an individual plant to being found and eaten by herbivores, known as an associational effect [1-5].

L46: I suggest adding an additional sentence between the ones that start and end on L46 to set up the idea that the neighborhood can either have a positive or negative effect on a given plant. The sentence beginning with 'When herbivores…' could be revised to start with 'For example, when herbivores…,' which would crea

---

## [Decision Letter · Decision Letter 1]

4 Aug 2025

Fine-scale associational effects: single plant neighbours can alter susceptibility of focal plants to herbivores

PONE-D-25-16194R1

Dear Dr. Finnerty,

We’re pleased to inform you that your manuscript has been judged scientifically suitable for publication and will be formally accepted for publication once it meets all outstanding technical requirements.

Kind regards,

Showkat Ahmad Ganie, Ph.D.

Academic Editor

PLOS ONE

Additional Editor Comments (optional):

The manuscript can be accepted for publication in PLOS one.

Reviewers' comments:

Reviewer's Responses to Questions

**Comments to the Author**

1. If the authors have adequately addressed your comments raised in a previous round of review and you feel that this manuscript is now acceptable for publication, you may indicate that here to bypass the “Comments to the Author” section, enter your conflict of interest statement in the “Confidential to Editor” section, and submit your "Accept" recommendation.

Reviewer #1: All comments have been addressed

Reviewer #2: All comments have been addressed

2. Is the manuscript technically sound, and do the data support the conclusions?

Reviewer #1: Yes

Reviewer #2: Yes

3. Has the statistical analysis been performed appropriately and rigorously? 

Reviewer #1: Yes

Reviewer #2: Yes

4. Have the authors made all data underlying the findings in their manuscript fully available?

Reviewer #1: Yes

Reviewer #2: Yes

5. Is the manuscript presented in an intelligible fashion and written in standard English?

Reviewer #1: Yes

Reviewer #2: Yes

6. Review Comments to the Author

Reviewer #1: (No Response)

Reviewer #2: All of my suggestions have been carefully and thoughtfully addressed. I appreciate the changes the authors made and I have no further comments.

7. PLOS authors have the option to publish the peer review history of their article (what does this mean?). If published, this will include your full peer review and any attached files.

Reviewer #1: No

Reviewer #2: No

---

## [Editor Report · Acceptance letter]

PONE-D-25-16194R1

PLOS ONE

Dear Dr. Finnerty,

I'm pleased to inform you that your manuscript has been deemed suitable for publication in PLOS ONE. Congratulations! Your manuscript is now being handed over to our production team.

Kind regards,

on behalf of

Dr. Showkat Ahmad Ganie

Academic Editor

PLOS ONE